# Association between age people started working and missing teeth in an elderly population in Ecuador: Evidence from a cross-sectional study

Camila Hallon[1], Camilo Barrionuevo-León[1], Juan Carlos Gallardo-Bastidas[2], Karla Robles-Velasco[1,3]*, Iván Cherrez-Ojeda[1,3]*, Marco Faytong-Haro[1,3,4]

1 School of Health, Universidad de Especialidades Espíritu Santo, Guayaquil, Ecuador, 2 Dentistry School, Universidad Católica Santiago de Guayaquil, Guayaquil, Ecuador, 3 Respiralab Research Group, Guayaquil, Ecuador, 4 Ecuadorian Development Research Lab, Daule, Guayas, Ecuador

* ivancherrez@gmail.com (ICO); karlaroblesvelasco@gmail.com (KRV)

## Abstract

Child labor has significant physical, psychological, and social consequences, which can persist into adulthood. This study investigates the association between the age at which an individual begins working and tooth loss in older adults in Ecuador. We analyzed data from the SABE 2009 survey (Survey of Health, Well-being, and Aging), using binary logistic regression to examine potential relationships. Our analytical sample comprised of 3,899 older adults from mainland Ecuador, with 42.50% having started working between the ages of 5 and 12. Unadjusted logistic regression results indicated that older adults who began working at ages 5–12 had a 42% higher risk of missing more than 4 teeth compared to those who started working at ages 18–25. After adjusting for potential confounders, the resulting risk was 28% higher than for the reference group [OR 1.28 95% CI 1.25–1.30]. Our findings demonstrate that early engagement in labor is a risk factor for tooth loss among older adults, displaying the long-term impacts of child labor on oral health. Health education and benefits should be provided to this vulnerable population for tooth loss prevention.

## Introduction

According to the International Labor Organization (ILO), child labor refers to work that is "mentally, physically, socially, or morally dangerous and harmful to children, and/or interferes with their schooling" [1]. In Ecuador, approximately 8.2% of the children aged 5 to 14 years old begin to work, whereas 8.9% of the children aged 7 to 14 work and study at the same time [2]. Approximately 89.9% work in the agriculture sector, 8.1% in services like domestic and street work, and 2% in industries including mining and construction. In the largest cities of Ecuador, Guayaquil, Quito, Cuenca, Machala and Ambato, child labor rates are 0.3%, 0.4%, 0.7%, 1.5%, and 3.4% respectively [3].

**Data Availability Statement:** The dataset file is available from https://www.ecuadorencifras.gob.ec/encuesta-de-salud-bienestar-del-adulto-mayor/.

**Funding:** The author(s) received no specific funding for this work.

**Competing interests:** The authors have declared that no competing interests exist.

Health in children is often recognized as the foundation for health in adults [4]. Child labor has been previously associated with stunting, wasting, and chronic malnutrition of working children [5], as well as delayed genital development in boys [6]. A 2014 study found that younger working children were more likely to report backaches, infections, burns and lung disease, whereas older working children had a higher probability of reporting exhaustion or tiredness [7]. Beyond the physical alterations, a 2012 study found that the prevalence of behavioral problems among 225 working children was approximately 9.8% [8].

Currently, there are no studies investigating the impact of child labor on the oral cavity, despite the evidence that psychological, social and environmental factors [9] can negatively impact oral health. Stress [10, 11] and social isolation [12, 13] are two well-documented adult examples, and, in children, poor dental health habits [14] and low education level [15] of the parents have been revealed to deteriorate oral health. In general, systemic health and oral health have shown a bidirectional relationship [16], which means preserving a healthy dentition is intrinsically linked to the quality of life of each individual [17].

The objective of this study was to explore the relationship between the age at which a sample of older adults began working and their number of teeth lost through logistic regression. To complete this objective, data from the 2009 Health, Well-Being and Aging Survey (SABE) was utilized, which is a household sampling survey of 5,235 older adults in 15 Ecuadorian provinces (the survey excludes the Amazon Rainforest and Galapagos Islands regions) collected by the National Institute of Statistics and Censuses (INEC) [18]. The questions asked include information about the health status, functional status, work history and use of medicines and services of the respondents, among other data.

Ecuador is a suitable country for exploring the impact of child labor on oral health for several reasons. First, as a developing nation, Ecuador faces various social, economic, and health challenges that make it particularly vulnerable to child labor [19]. Investigating the influence of child labor on oral health in Ecuador can provide valuable insights into the long-term consequences of such practices in similar low-income and middle-income countries.

Second, the oral healthcare system in Ecuador, as in many other non-industrialized countries, faces limitations in terms of access, resources, and awareness [20]. By studying the impact of child labor on oral health in Ecuador, researchers can identify specific needs and develop targeted interventions to improve oral health outcomes in vulnerable populations.

Finally, studying the child labor influence on oral health in Ecuador can contribute to a better understanding of the relationship between child labor and oral health in the broader context of the social determinants of health. This can help inform policies and interventions aimed at addressing child labor, improving oral health, and promoting overall well-being in Ecuador and other similar settings.

## Materials and methods

### Study design and setting

In this study, we employed a cross-sectional design, utilizing data from the national 2009 Survey of Health, Well-Being, and Aging (SABE). The SABE survey is a household-based sample survey that investigates the older adult population considered adults aged 60 and above in 15 Ecuadorian provinces, excluding the Amazon Rainforest and Galapagos regions. Conducted by the National Institute of Statistics and Census' (INEC) Department of Sociodemographic Statistics, the survey aimed to assess demographic characteristics, cognitive development, health status, medication usage, and service accessibility among other data of approximately 5235 older adults interviewed between June and August 2009. INEC carried out a multistage probabilistic sampling, first by areas within the provinces and then by households, in which

the units of analysis or observation were selected at random to ensure that each element had a probability of selection different to zero. To select the interviewee, the questioner had a map, sketch or list where the homes they should visit were marked. This process ensured the representativeness of the sample in the selected provinces. The present study utilizes publicly available open data and, as such, did not require Institutional Review Board (IRB) approval. Research of this type, involving open and publicly accessible data, is typically considered exempt from review according to prevailing Ecuadorian ethical guidelines.

## Participants

The study selectively focused on older adults who had engaged in employment at least once throughout their lifespan, with the aim of investigating the impact of early work commencement on oral cavity conditions.

## Variables and measurement

**Outcome variable.** The outcome variable chosen to determine the oral health status of the participants was their number of teeth missing at the time of the survey. Tooth loss is often considered the most useful oral health indicator, as it portrays the performance of the individual and the dentist regarding oral hygiene, the individual's access to dental services and their cultural beliefs [21]. The original SABE question utilized to create this variable was: "Now, I would like to ask you some questions about your mouth and teeth. Please tell me, are you missing any teeth or molars?" [18]. The categories for interviewees' answers consisted of: "Yes, a few (up to 4 teeth)", "Yes, a lot (more than 4 and less than half)", "Yes, the majority (half or more)", "Yes, all are missing", "No", "Doesn't know" and "Doesn't answer". For this reason, in the present study we chose to divide the respondents into those who had not lost any or up to 4 teeth, and those who were missing more than 4 teeth or all. "Doesn´t know" and "Doesn´t answer" responses were subsequently eliminated.

**Predictor variable.** Due to the mental and physical stress that child labor produces on a child, we chose the age at which the participants started working for the first time as our predictor variable for tooth loss. The exact SABE question utilized was: "How old were you when you first started working?" [18]. For this question, we classified all ages responded into five categories: 5 to 12 years old, 13 to 17 years old, 18 to 25 years old, 26 to 35 years old, and 36 to 80 years old. People who did not know or did not answer were not taken into account. The age ranges for this variable were established based on significant educational and social milestones in Ecuador, as well as the distribution of the sample data. The 5 to 12 age group corresponds to children finishing primary school, an age at which some might be forced to work due to socioeconomic factors. The range of 13 to 17 years summarizes the period before completing secondary education, with 18 years being the age at which one is no longer considered a minor and is expected to have a high school diploma, a basic requirement for many jobs. Most study participants began working at age 25, as evidenced by the 90th percentile. The 26 to 35 age range in Ecuador captures those who complete extended education, who take longer to obtain their degrees and those who choose to start working in this period without prior formal studies. The decision to have a broader age range, from 36 to 80 years, is due to the fact that they constitute a smaller portion of the sample, only 2.72%, so combining them provides an optimal analysis without compromising the integrity of the data.

**Control variables.** Control variables were selected to isolate the relationship between the outcome and predictor variables from other potentially influential factors. These control variables came from the questions within the same SABE 2009 survey, located in the sections of personal data (section A), health status (section C), use and accessibility of services (section F)

and work history (section H). These control variables covered demographic characteristics including age, sex, education, ethnicity, marital status, and living arrangements (living alone or accompanied), diseases including diabetes, osteoporosis, cancer, and nervous or mental disorders, as well as other important data like tobacco use, childhood socioeconomic situation, childhood period where food was scarce and the participant felt hungry, and type of insurance.

For the age variable, participants were categorized based on their age at the time of the survey into the groups 60 to 65 years old, 66 to 70 years old, 71 to 75 years old, 76 to 80 years old and 81+ years old based off the idea that oral health commonly declines with ageing [22]. They were also classified by sex as male or female, considering the differences in physiology that exist between each [23]. Participant's education was also classified into levels such as no education, primary education, secondary education, and postsecondary education, due to its close relationship with socioeconomic status [24].

Participants were also categorized on the basis of their self-reported ethnicity as mestizo, black, white, or indigenous, since these groups may have different access to dental services or habits that should be kept in mind [25]. For marital status, participants were classified into two groups: those who had been married or partnered at least once in their lives, and those who had never been married or partnered. There are currently several studies that connect marital status with general health [26, 27], which is why we chose to include it as a control variable. Living arrangements were categorized based on whether participants lived alone or accompanied [28], since functional independence is usually lost with ageing [28].

Diagnoses of various illnesses–including diabetes [29], cancer [30], osteoporosis [31] and nervous or mental disorders [32] were also included as control variables since all of them have previously shown unique connections to oral health. The study's participants were also classified into those who smoke or smoked cigarettes and those who have never smoked cigarettes in their lifetime, due to the negative association smoking possesses with oral health [33]. Two specific questions were included to further delineate the impact of childhood socioeconomic status and malnutrition on oral health. First, participants were asked to characterize their childhood socioeconomic situation as 'bad', 'good', or 'regular'. Second, they were asked whether they experienced periods of food scarcity and hunger during their childhood, with response options being 'yes' or 'no'. These variables aim to isolate, as much as possible, the deteriorative effects of malnutrition on the oral cavity [34]. Finally, participants were divided based on their owned health insurance into public insurance, private insurance or no insurance, to analyze the efficacy of the health systems within Ecuador.

## Study size and missing data

In our study, the original sample was made up of 5235 older adults over 60 years of age in 15 provinces of continental Ecuador. A total of 569 participants were excluded due to their lack of response to the question regarding the age at which they started working, our predictor variable. An additional 367 subjects were omitted because they did not know or did not provide their age at the time of the survey. Additionally, 8 participants who were missing information on their educational level, and an extra 177 who did not mention their self-perceived ethnicity, were eliminated. 64 participants who did not respond to the question "Has a doctor or nurse ever told you if you have diabetes, that is, high blood sugar levels?" were eliminated, and 3 respondents were eliminated in the same way who did not respond about whether or not they have had cancer in their lives. Additionally, 41 responses were eliminated for not answering whether a doctor had previously diagnosed them with osteoporosis or not, and 8 responses were eliminated for not answering whether a doctor had previously diagnosed them with a nervous or mental disorder. 7 responses were eliminated because the participants did not

answer whether they smoked or smoke cigarettes, 22 responses because they did not mention their childhood socioeconomic situation, and 66 responses because they did not know or did not respond if food was ever scarce and they felt hungry in their childhood. 4 answers were deleted because the participants did not know what kind of insurance they had. After accounting for these missing values, our final analytic sample included 3899 older adults.

### Statistical methods

Initial descriptive analyses summarized the distribution of the study variables using percentages. Cross-tabulations examined the bivariate relationships between the outcome variable and each of the predictor and control variables, which can be found in Table 1. The Pearson Chi-square test assessed the statistical significance of these associations, with P-values < 0.05 deemed statistically significant.

Logistic regression analysis was performed to test the association between starting work age and tooth loss. Table 2 shows the odds ratio and standard errors of the main 101 explanatory variables and controls. Model 1 represents the unadjusted model, and Model 2 represents the adjusted model with all mentioned controls. In this study, weights were also applied to Model 2 to make the findings representative of the study population.

For sensitivity purposes, we conducted the main regression incorporating all control variables, utilizing an alternative categorization for the age of work initiation, and without applying the weights. These alternative specifications can be found in the supplementary materials, where both tables are provided. Both specifications did not alter the main findings. Importantly, all models—both main and sensitivity analyses—were performed with robust standard errors to ensure the reliability of our results.

## Results

### Descriptive results

Approximately 12% had a maximum of four or no missing teeth. 47% of our sample were female. Almost a third of the sample was aged 60 to 65 years old and 22% were 66 to 70 years old. About 61% had only primary education, and one-quarter had no education at all. Most participants considered themselves to be mestizos (73.48%), followed by 13% white, 10% indigenous, and 3% black. Ninety-six percent of the participants were married or partnered at least once. Approximately 10% of the patients live alone.

About 12% of our sample had been diagnosticated with diabetes. 3% with cancer and 17% with osteoporosis. Approximately 11% had a nervous or mental disorder diagnosis. Almost half of the sample smoked or smoke cigarettes (43%). Most participants when asked described their childhood socioeconomic as regular (41%), and around 22% reported it as bad. Thirty-five percent of the sample reported a childhood period where food was scarce, and they felt hungry. 69% had no insurance, and only 2% had private insurance.

Respondents who started working at ages 5 to 12 years old were more likely to be part of the group with more than four missing teeth (p = 0.083). There were more females with more than four missing teeth (p = 0.000). Having less education meant a larger proportion of patients with more than four missing teeth (p = 0.000). People considered mestizos were more likely to be in the group with more than four missing teeth, although the differences were negligible (p = 0.676). The same happened to people who were never married (p = 0.731) or partnered, live alone (p = 1.121), and have diabetes (p = 1.120).

Participants with a cancer diagnosis were more frequently observed in the category of those missing more than four teeth (p = 0.062). Participants with no insurance had a larger proportion of people missing more than four teeth (p = 0.000). People who reported having a

**Table 1.  Descriptive statistics of model variables stratified by total and outcome variable (n = 3899).**

| Variables | Missing up to 4 teeth or none | Missing more than 4 teeth or all | P-value | Row total |
|---|---|---|---|---|
| **Age when first started working (predictor)** | | | | |
| 5–12 | 38.44% | 43.04% | 0.083 | 42.50% |
| | [34.00% - 42.89%] | [41.39% - 44.70%] | | |
| 13–17 | 29.37% | 29.74% | | 29.70% |
| | [25.21% - 33.54%] | [28.21% - 31.27%] | | |
| 18–25 | 25.05% | 19.82% | | 20.44% |
| | [21.09% - 29.02%] | [18.49% - 21.15%] | | |
| 26–35 | 4.97% | 4.60% | | 4.64% |
| | [2.98% - 6.95%] | [3.90% - 5.30%] | | |
| 36–80 | 2.16% | 2.79% | | 2.72% |
| | [0.83% - 3.49%] | [2.24% - 3.35%] | | |
| **Sex** | | | | |
| Female | 35.42% | 48.11% | 0.000 | 46.60% |
| | [31.05% - 39.79%] | [46.43% - 49.78%] | | |
| Male | 64.58% | 51.89% | | 53.40% |
| | [60.21% - 68.95%] | [50.22% - 53.56%] | | |
| **Age** | | | | |
| 60–65 | 53.56% | 28.61% | 0.000 | 31.57% |
| | [49.00% - 58.12%] | [27.10% - 30.12%] | | |
| 66–70 | 23.33% | 22.21% | | 22.34% |
| | [19.46% - 27.19%] | [20.82% - 23.60%] | | |
| 71–75 | 12.74% | 19.21% | | 18.44% |
| | [9.69% - 15.79%] | [17.89% - 20.53%] | | |
| 75–80 | 6.26% | 14.81% | | 13.80% |
| | [4.05% - 8.48%] | [13.63% - 16.00%] | | |
| 81+ | 4.10% | 15.16% | | 13.85% |
| | [2.29% - 5.92%] | [13.96% - 16.36%] | | |
| **Educational level** | | | | |
| No education | 17.06% | 24.68% | 0.000 | 23.78% |
| | [13.62% - 20.50%] | [23.24% - 26.12%] | | |
| Primary | 58.96% | 61.58% | | 61.27% |
| | [54.47% - 63.46%] | [59.96% - 63.21%] | | |
| Secondary | 12.74% | 9.95% | | 10.28% |
| | [9.69% - 15.79%] | [8.95% - 10.95%] | | |
| Postsecondary | 11.23 | 3.78% | | 4.67% |
| | [8.34% - 14.12%] | [3.15% - 4.42%] | | |
| **Ethnic self-report** | | | | |
| Mestizo | 72.35% | 73.63% | 0.068 | 73.48% |
| | [68.27% - 76.44%] | [72.16% - 75.11%] | | |
| Black | 4.10% | 3.41% | | 3.49% |
| | [2.29% - 5.92%] | [2.80% - 4.01%] | | |
| White | 12.10% | 12.83% | | 12.75% |
| | [9.11% - 15.07%] | [11.72% - 13.95%] | | |
| Indigenous | 11.45% | 10.13% | | 10.28% |
| | [8.54% - 14.36%] | [09.12% - 11.13%] | | |
| **Marital Status** | | | | |

*(Continued)*

**Table 1.** (Continued)

| Variables | Missing up to 4 teeth or none | Missing more than 4 teeth or all | P-value | Row total |
|---|---|---|---|---|
| Never married or partnered | 4.10% | 4.45% | 0.731 | 4.41% |
| | [2.29% - 5.92%] | [3.76% - 5.14%] | | |
| Married or partnered now or before | 95.90% | 95.55% | | 95.59% |
| | [94.08% - 97.71%] | [95.86% - 96.24%] | | |
| **Living arrangement** | | | | |
| Lives accompanied | 91.79% | 89.46% | 0.121 | 89.74% |
| | [89.28% - 94.30%] | [88.44% - 90.49%] | | |
| Lives alone | 8.21% | 10.54% | | 10.26% |
| | [5.70% - 10.72%] | [9.51% - 11.56%] | | |
| **Diabetes** | | | | |
| No | 89.85% | 87.31% | 0.120 | 87.61% |
| | [87.09% - 92.61%] | [86.20% - 88.42%] | | |
| Yes | 10.15% | 12.69% | | 12.39% |
| | [7.39% - 12.91%] | [11.58% - 13.80%] | | |
| **Cancer** | | | | |
| No | 98.70% | 97.24% | 0.062 | 97.41% |
| | [97.67% - 99.74%] | [96.69% - 97.78%] | | |
| Yes | 1.30% | 2.76% | | 2.59% |
| | [0.26% - 2.33%] | [2.22% - 3.31%] | | |
| **Osteoporosis** | | | | |
| No | 85.75% | 83.00% | 0.137 | 83.33% |
| | [82.55% - 88.94%] | [81.75% - 84.26%] | | |
| Yes | 14.25% | 17.00% | | 16.67% |
| | [11.06% - 17.45%] | [15.074% - 18.25%] | | |
| **Nervous or mental disorder** | | | | |
| No | 90.50% | 89.32% | 0.438 | 89.46% |
| | [87.82% - 93.18%] | [88.29% - 90.35%] | | |
| Yes | 9.50% | 10.68% | | 10.54% |
| | [6.82% - 1.22%] | [9.65% - 11.71%] | | |
| **Smoke or smoked cigarettes** | | | | |
| No | 57.45% | 56.43% | 0.678 | 56.55% |
| | [52.93% - 61.97%] | [54.77% - 58.09%] | | |
| Yes | 42.55% | 43.57% | | 43.45% |
| | [38.03% - 47.07%] | [41.91% - 45.23%] | | |
| **Childhood socioeconomic status** | | | | |
| Bad | 21.17% | 22.53% | 0.745 | 22.36% |
| | [17.43% - 24.90%] | [21.13% - 23.92%] | | |
| Good | 36.93% | 37.14% | | 37.11% |
| | [32.52% - 41.35%] | [35.52% - 38.75%] | | |
| Regular | 41.90% | 40.34% | | 40.52% |
| | [37.39% - 46.41%] | [38.69% - 41.98%] | | |
| **Childhood period where food was scarce and you felt hungry** | | | | |
| No | 70.41% | 64.76% | 0.016 | 65.43% |
| | [66.24% - 74.58%] | [63.16% - 66.35%] | | |
| Yes | 29.59% | 35.24% | | 34.57% |
| | [25.41% - 33.76%] | [33.65% - 36.84%] | | |
| **Health insurance type*** | | | | |

(Continued)

**Table 1.** (Continued)

| Variables | Missing up to 4 teeth or none | Missing more than 4 teeth or all | P-value | Row total |
|---|---|---|---|---|
| No insurance | 60.91% | 69.18% | 0.000 | 68.20% |
| | [56.45% - 65.37%] | [67.63% - 70.72%] | | |
| Private | 4.75% | 2.10% | | 2.41% |
| | [2.81% - 6.70%] | [1.62% - 2.57%] | | |
| Public only | 34.34% | 28.73% | | 29.39% |
| | [30.00% - 38.68%] | [27.21% - 30.24%] | | |

* Total is not a 100% because participants could have both private and public insurances under the private insurance category.

Confidence intervals are in square brackets.

childhood period where food was scarce, and they felt hungry were more likely to be missing more than four teeth (p = 0.016).

### Regression results

Model 1, the unadjusted logistic regression model, was initially employed to examine the relationship between starting work at different ages and the risk of missing more than four teeth in older adults. In this model, only the age at which individuals started working was considered without accounting for other control variables. Model 1 showed the association between starting to work at ages 5 to 12 years old and the risk of losing more than 4 teeth had an Odds Ratio (OR) equal to 1.42 [95% CI 1.10–1.82]. This finding suggests that older adults who began working at 5–12 years old have a 42% higher risk of tooth loss than those who started working between the ages of 18 and 25 years old. The risk was also higher for individuals who started working between the ages of 13–17 [OR 1.28, 95% CI 0.98–1.67], although it was less pronounced in comparison to the 5–12 age group.

Model 2 not only included the main predictor variable, but also incorporated numerous control variables to account for other factors that might be associated with missing more than four teeth. These control variables were sex, age, educational level, ethnic self-report, marital or partnership status, living arrangement, diagnosis of diabetes, cancer, osteoporosis, nervous or mental disorders, tobacco use, childhood socioeconomic status, childhood period where food was scarce and the participant felt hungry, and health insurance type. The fully adjusted logistic regression model demonstrated an association between starting work at ages 5–12 and missing more than 4 teeth, with an OR of 1.28 [95% CI 1.25–1.30].

When comparing age groups, the risk of missing more than 4 teeth increased as participants were older. Participants who were 66 to 70 years old had a 75% higher risk of missing more than four teeth [OR 1.75, 95% CI 1.72–1.78] than those who were 60 to 65 years old, and those who were 81+ years old had the highest tooth loss risk of all age groups [OR 6.69, 95% CI 6.48–6.91]. Gender also played a role in tooth loss, as men had an OR of 0.46 [95% CI 0.46–0.47], indicating a 54% lower risk of losing more than 4 teeth when compared to women.

Model 2 revealed the significant impact of education level on tooth loss. Individuals with postsecondary education had an OR of 0.53 [95% CI 0.52–0.55], which indicates a lower risk of missing more than four teeth compared to those without education. This finding suggests that higher educational levels contribute to better oral health outcomes.

Ethnic self-reporting was also considered in the model. Black individuals had an OR of 0.83 [95% CI 0.80–0.86], showing a lower risk of missing more than four teeth than those of mestizo ethnicity. For white individuals, the risk for tooth loss was 18% lower than for mestizos [OR 0.82, 95% CI 0.80–0.83].

**Table 2. Odds ratio from logistic regression models predicting missing teeth (more than 4).**

| Variables | Model 1: Missing more than 4 teeth | Model 2: Missing more than 4 teeth |
|---|---|---|
| **Age when first started working = 1, 5–12** | 1.415*** | 1.277*** |
| | (SE 0.181) | (SE 0.0125) |
| | [95% CI 1.101–1.819] | [95% CI 1.253–1.302] |
| **Age when first started working = 2, 13–17** | 1.280* | 1.182*** |
| | (SE 0.174) | (SE 0.0117) |
| | [95% CI 0.981–1.670] | [95% CI 1.159–1.205] |
| **Age when first started working = 4, 26–35** | 1.170 | 0.669*** |
| | (SE 0.286) | (SE 0.0104) |
| | [95% CI 0.724–1.890] | [95% CI 0.649–0.690] |
| **Age when first started working = 5, 36–80** | 1.635 | 1.305*** |
| | (SE 0.568) | (SE 0.0304) |
| | [95% CI 0.828–3.229] | [95% CI 1.247–1.366] |
| **Sex = Male** | | 0.464*** |
| | | (SE 0.00418) |
| | | [95% CI 0.456–0.472] |
| **Age = 2, 66–70** | | 1.751*** |
| | | (SE 0.0147) |
| | | [95% CI 1.722–1.780] |
| **Age = 3, 71–75** | | 2.767*** |
| | | (SE 0.0293) |
| | | [95% CI 2.710–2.825] |
| **Age = 4, 76–80** | | 5.335*** |
| | | (SE 0.0796) |
| | | [95% CI 5.181–5.493] |
| **Age = 5, 81+** | | 6.692*** |
| | | (SE 0.107) |
| | | [95% CI 6.485–6.905] |
| **Educational level = 1, Primary** | | 0.959*** |
| | | (SE 0.00971) |
| | | [95% CI 0.941–0.979] |
| **Educational level = 2, Secondary** | | 0.902*** |
| | | (SE 0.0122) |
| | | [95% CI 0.878–0.926] |
| **Educational level = 3, Postsecondary** | | 0.533*** |
| | | (SE 0.00831) |
| | | [95% CI 0.517–0.550] |
| **Ethnic self-report = 2, Black** | | 0.829*** |
| | | (SE 0.0154) |
| | | [95% CI 0.799–0.859] |
| **Ethnic self-report = 3, White** | | 0.818*** |
| | | (SE 0.00825) |
| | | [95% CI 0.802–0.835] |
| **Ethnic self-report = 4, Indigenous** | | 0.777*** |
| | | (SE 0.00925) |
| | | [95% CI 0.759–0.795] |
| **Marital status = Married or partnered now or before** | | 0.665*** |

*(Continued)*

**Table 2.** (Continued)

| Variables | Model 1: Missing more than 4 teeth | Model 2: Missing more than 4 teeth |
|---|---|---|
| | | (SE 0.0132) |
| | | [95% CI 0.640–0.691] |
| **Lives alone = Lives alone** | | 1.084*** |
| | | (SE 0.0136) |
| | | [95% CI 1.057–1.111] |
| **Has diabetes = Yes** | | 1.764*** |
| | | (SE 0.0205) |
| | | [95% CI 1.725–1.805] |
| **Has cancer = Yes** | | 1.551*** |
| | | (SE 0.0377) |
| | | [95% CI 1.479–1.626] |
| **Has osteoporosis = Yes** | | 0.864*** |
| | | (SE 0.00816) |
| | | [95% CI 0.848–0.880] |
| **Has a nervous or mental disorder = Yes** | | 1.112*** |
| | | (SE 0.0129) |
| | | [95% CI 1.087–1.138] |
| **Smoke or smoked cigarettes = Yes** | | 1.576*** |
| | | (SE 0.0128) |
| | | [95% CI 1.551–1.601] |
| **Childhood socioeconomic situation = 1, Good** | | 1.192*** |
| | | (SE 0.0121) |
| | | [95% CI 1.168–1.216] |
| **Childhood socioeconomic situation = 2, Regular** | | 1.129*** |
| | | (SE 0.0108) |
| | | [95% CI 1.108–1.150] |
| **Childhood period where food was scarce and you felt hungry = Yes** | | 1.258*** |
| | | (SE 0.0102) |
| | | [95% CI 1.238–1.279] |
| **Insurance type = 2, Private** | | 0.677*** |
| | | (SE 0.0111) |
| | | [95% CI 0.656–0.699] |
| **Insurance type = 3, Public only** | | 0.857*** |
| | | (SE 0.00670) |
| | | [95% CI 0.844–0.871] |
| **Constant** | 5.871*** | 6.214*** |
| | (0.590) | (0.151) |
| **Observations** | 3,899 | 3,899 |

Standard errors in parentheses

95% Confidence intervals in brackets

*** p<0.01

** p<0.05

* p<0.1

Marital or partnership status and living arrangements were also included as control variables. The results indicate that those who were married or partnered at least once had a lesser risk of losing more than 4 teeth [OR 0.67, 95% CI 0.64–0.69], and those who live alone had a 8% higher risk of missing more than 4 teeth [OR 1.08, 95% CI 1.06–1.11].

Participants who had been diagnosticated with diabetes had a 76% higher risk of missing more than four teeth [OR 1.76, 95% CI 1.73–1.81]. Those with cancer had an OR of 1.55 [95% CI 1.48–1.63]. Nervous or mental disorders increased the risk for tooth loss [OR 1.11, 95% CI 1.09–1.14]. Former smokers and current smokers had a higher risk for tooth loss [OR 1.58, 9% CI 1.55–1.60]. Participants who answered yes to having a childhood period where food was scarce and they felt hungry had a 25% higher risk of missing more than 4 teeth [OR 1.26, 95% CI 1.24–1.28].

Overall, Model 2, which was fully adjusted for various demographic and social factors and illnesses, confirmed an association between starting work at a young age (5–12) and an increased risk of tooth loss in older adults. This highlights the importance of considering early work experiences and other social determinants in understanding oral health outcomes in older populations.

## Sensitivity analysis

Non-weighted sensitivity analysis to confirm our main findings was performed with the age classification of 5 to 9 years old, 10 to 19 years old, 20 to 59 years old and 60 to 80 years old (S1 Table). This stratification, which segments the data into childhood, adolescence, adulthood, and senior adulthood, allows for a thorough examination of the influences at each distinctive life phase. In the adjusted model, participants who started working at 5 to 9 years old had a 55% higher risk of missing more than 4 teeth in comparison to those who started working at 20 to 59 years old [OR 1.55, 95% CI 1.07–2.25]. Similarly, those who started working at 10 to 19 years old had a 30% higher risk for tooth loss [OR 1.30, 95% CI 1.00–1.69]. The results obtained demonstrate starting work at an early age is a risk factor for tooth loss. In S2 Table, we conduct the primary analysis utilizing the original classifications for work start age, yielding results that exhibit only trivial variations. In analyses not displayed, we also executed a sensitivity analysis employing weights with the alternative age classification. The outcomes of this weighted analysis did not significantly alter the results, affirming the stability and robustness of our primary findings under various methodological conditions.

## Discussion

In the present study the results of the adjusted logistic regression suggest that people who started working between the ages of 5 to 12 years old have a 28% higher risk [OR 1.28 95% CI 1.25–1.30] of losing more than 4 teeth as older adults than those who started working between the ages of 18 to 25 years. Additionally, sex, educational level, smoking, childhood period where food was scarce, and diagnosis of diabetes and/or cancer were found to increase the risk of losing more than 4 teeth in an individual.

Keles et al. in 2018 analyzed oral health status and its implications for quality of life in a sample of 514 adolescents who were working and studying at the same time [35]. They found that approximately 91.6% of the participants had diseased periodontal tissues, and 43.4% had never visited a dentist. 11.5% of the sample did not brush their teeth. Adolescent oral health has different needs to be addressed due to a commonly high caries rate, an increased risk of traumatic injury, and an increased aesthetic desire and awareness [36]. Broadbent et al. described in their 2006 study that adults who held favorable dental beliefs as adolescents had fewer teeth lost due to dental caries and periodontal disease than those who did not [37].

Current literature describes that approximately 30% of children under 7 years of age suffer dental trauma to one or more primary incisors, and about 40% visit the dentist for the first time for this reason [38]. Recalling the previously mentioned information, working children are more prone to suffer accidents during working hours [7]. As a consequence of falls or blows, children may lose or require endodontic treatment on affected deciduous teeth, and experience sequelae on their permanent teeth. The most common are enamel discoloration and hypoplasia [39]. The negative impact of dental trauma on the quality of life of children has been proven several times before [40, 41], so it is essential to inform caregivers how to act in the face of these events.

The control variables applied in this study revealed that women have a higher risk of tooth loss than men, a finding congruent with current literature. For example, research by Mundt et al. in 2007 found that women have a higher risk of tooth loss than men, especially when they are in unfavorable psychosocial situations such as unemployment or lack of access to education [42]. Similarly, Meisel et al. found in 2008 that women had a greater number of missing teeth in contrast to men, even though they had better overall periodontal health. They attributed this finding to an increased rate of bone turnover during pregnancy, and unfavorable socioeconomic conditions such as low education and low social status [43].

In our study, people who identified as black had a lower probability of losing 4 or more teeth than people who identified as mestizo [OR 0.83, 95% CI 0.80–0.86]. This finding contrasts with a 2018 study that used the SABE survey to investigate oral health in Brazil, and showed that those who identified as black and mulatto were respectively 65% and 45% less likely to have a functional dentition (more than 21 teeth present) [44]. Additionally, in a study published in 2015 based on the National Health and Nutrition Examination (NHANES) survey, non-Hispanic black adults showed less tooth retention (38%) when compared to other races (Non-Hispanic white, non-Hispanic Asian, Hispanic) [45].

Overall, higher education showed a protective effect against tooth loss within our results. Those with postsecondary education had a 47% lesser risk of losing more than 4 teeth than those with no education [OR 1.53, 95% CI 0.52–0.55]. A SABE survey-based study published in 2020 showed older adults in Brazil had a 96% lesser risk of being edentulous when they had 8 or more years of schooling [46]. Another 2015 study that utilized SABE survey results of 7 Latin American and Caribbean cities found that people with higher education reported fewer teeth lost [47].

Comparing age groups within our study, those who were 81 years old or older had the highest probability of losing more than 4 teeth in relation to the participants aged 60 to 65 years old [OR 6.69, 95% CI 6.48–6.91]. Following them were people aged 76 to 80 [OR 5.33 95% CI 5.18–5.49], 71 to 75 [OR 2.77 95% CI 2.71–2.82], and finally people aged 66 to 70 [OR 1. 75% CI 1.72–1.78]. Despite the possibility that this is a result of chance, a 2019 study found that the frequency of dental visits declines with age due to immutable factors such deteriorating general health [48]. This fact is particularly significant in light of the fact that older persons are more likely to get oral infections, particularly if they reside in a long-term care facility or have dementia [49]. The elevated risk for tooth loss may also be explained by a decrease in functional independence [28].

Regarding smoking, participants who smoke or smoked had a higher risk of losing more than 4 teeth than those who did not [OR 1.58 95% CI 1.55–1.60]. Similarly, 2021 study based on the Geriatric Oral Health Assessment Index (GOHAI) of Colombia found that a higher exposure to tobacco was related to a lower number of remaining teeth [50]. There is abundant evidence that smoking is an important risk factor for periodontitis and tooth loss [51, 52], which can explain these findings. In addition to smoking, diabetes illness is another established

risk factor for periodontitis [53], and in this study, diabetics' risk of losing more than four teeth was 76% [OR 1.76 95% CI 1.72–1.81] higher than that of people without the condition.

Finally, among individuals who said they did not have enough food to consume during a childhood period, the probability of losing more than four teeth was 26% greater [OR 1.26 95% CI 1.24–1.28]. This variable was added because it was the most accurate representation of the participant's nutrition as a child. Guerrero et al. published a study in 1978 in which they examined 2 groups of 140 children aged 6 to 12 years old from different socioeconomic levels, and found that tooth development and eruption were delayed in the group of undernourished children [54]. The same discovery had the 2017 study made by Da Fonseca, where he described that inadequate nutrition deprives children of nutrients necessary for growth and development, including that of oral structures [55]. Lastly, a more recent study, published in 2022, found an association between malnutrition in children and the occurrence of caries lesions in permanent teeth [56].

Knowing whether there is a greater risk for tooth loss is important because tooth loss leads to functional impairment that is exacerbated when more teeth are lost [57], and has repercussions in other aspects like diet and self-esteem. In other words, from a dental standpoint, it is crucial to preserve an adequate number of functional teeth since the more teeth that are missing, the greater chewing and phonetic challenges are [58]. Humans need at least 3 to 4 occlusal units of posterior teeth (occluding premolars represent 1 occlusal unit, while occluding molars represent 2 occlusal units) for occlusal stability [59], whilst anterior teeth are often necessary to produce certain sounds including "p," "b", "f","v","th"" and "s" [60]. A trained dentist can rehabilitate fully or partially edentulous mouths using different appliances; nonetheless, not everyone has access to such resources. According to the World Health Organization [61], most oral diseases are preventable through modifiable risk factors including the consumption of sugar, tobacco, alcohol and poor hygiene. For this reason, appropriate measures should be implemented in our country to reduce edentulism by attacking its most common causes: dental caries and periodontal disease.

Among the limitations of this study, a major drawback is the age of the SABE 2009 survey. Furthermore, because the SABE Survey was conducted in only 15 Ecuadorian provinces, excluding the Amazon Rainforest and Galapagos regions, the findings of our study may not be generalizable to these unrepresented regions. Reliance on self-reported data may have introduced recall and social desirability biases into our variables, and also resulted in missing data that reduced our analytical sample. Another limitation is that our study, due to its cross-sectional design, can only determine the association between age at work start and tooth loss, not whether child labor causes tooth loss. Finally, our study could be subject to confounding bias due to unobserved variables that may influence tooth loss.

## Conclusion

Within the results of this cross-sectional study, it was observed that there is a relationship between starting to work at an early age and tooth loss. Older adults who began working between 5 and 12 years of age had a 28% higher risk of missing more than 4 teeth than those who began working between 18 and 25 years of age, while those who began working between 13 and 17 years of age had a 18% higher risk. The information presented could be considered a risk factor in the future with support from other investigations. If so, health care policies should be modified accordingly to reduce the risk of tooth loss in this vulnerable group through proper health education and benefits. More research should be conducted to understand the effects of child labor on adult oral health, especially in non-industrialized countries such as Ecuador.

## Supporting information

**S1 Table. Non-weighted sensitivity analysis of predictor and control variables.** S1 Table showcases a non-weighted sensitivity analysis of predictor and control variables related to the outcome of "missing more than 4 teeth" across two models: a bivariate model (Model 1) and a model incorporating all controls (Model 2). The table provides odds ratios, standard errors (SE), and 95% confidence intervals (CI) for various demographic, health, and lifestyle factors, including age, sex, educational level, ethnicity, marital status, health conditions, smoking status, childhood socioeconomic situation, and insurance type, with data drawn from a sample of 3,899 individuals.
(DOCX)

**S2 Table. Main analysis, non-weighted.** S2 Table illustrates a non-weighted main analysis of predictor and control variables related to the outcome of "missing more than 4 teeth" across two models: a bivariate model (Model 1) and a comprehensive model with all controls (Model 2). The table provides odds ratios, standard errors (SE), and 95% confidence intervals (CI) for a variety of demographic, health, and lifestyle factors, including age at which work began, sex, educational attainment, ethnicity, marital status, living situation, medical conditions, smoking habits, childhood socioeconomic conditions, and insurance type. The data is sourced from a sample of 3,899 individuals, with significance levels indicated by asterisks.
(DOCX)

## Author Contributions

**Conceptualization:** Camila Hallon, Camilo Barrionuevo-León, Iván Cherrez-Ojeda.

**Formal analysis:** Camila Hallon, Camilo Barrionuevo-León, Marco Faytong-Haro.

**Methodology:** Camila Hallon, Camilo Barrionuevo-León.

**Project administration:** Iván Cherrez-Ojeda, Marco Faytong-Haro.

**Resources:** Juan Carlos Gallardo-Bastidas, Iván Cherrez-Ojeda.

**Supervision:** Marco Faytong-Haro.

**Visualization:** Camila Hallon, Camilo Barrionuevo-León.

**Writing – original draft:** Camila Hallon, Camilo Barrionuevo-León.

**Writing – review & editing:** Camila Hallon, Camilo Barrionuevo-León, Karla Robles-Velasco, Iván Cherrez-Ojeda, Marco Faytong-Haro.

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
