## [Decision Letter · Decision Letter 0]

31 Aug 2023

PONE-D-23-21090Association between age people started working and missing teeth in an elderly population in Ecuador: evidence from a cross-section studyPLOS ONE

Dear Dr. Cherrez-Ojeda,

Thank you for submitting your manuscript to PLOS ONE. After careful consideration, we feel that it has merit but does not fully meet PLOS ONE’s publication criteria as it currently stands. Therefore, we invite you to submit a revised version of the manuscript that addresses the points raised during the review process.

We look forward to receiving your revised manuscript.

Kind regards,

Ana Cristina Mafla

Academic Editor

PLOS ONE

Journal Requirements:

2. English editions are required and please refer to the Data Availability Statement in the manuscript.

Reviewers' comments:

Reviewer's Responses to Questions

**Comments to the Author**

1. Is the manuscript technically sound, and do the data support the conclusions?

Reviewer #1: Partly

Reviewer #2: Yes

2. Has the statistical analysis been performed appropriately and rigorously? 

Reviewer #1: Yes

Reviewer #2: No

3. Have the authors made all data underlying the findings in their manuscript fully available?

Reviewer #1: Yes

Reviewer #2: No

4. Is the manuscript presented in an intelligible fashion and written in standard English?

Reviewer #1: Yes

Reviewer #2: No

5. Review Comments to the Author

Reviewer #1: Abstract

Appropriate for the text, I think it should give more strength to the conclusion.

Introduction

The introduction is adequate, it leads you to the knowledge gap that you want to solve with the article, however, it leaves out some confusing factors that are definitely key in the process of understanding early work with oral health conditions. This is how aspects such as the social determinants of health, which are so important to give strength to the relationship with early work, are not taken into account to give strength to this analysis. It is a good initial approximation to the subject but it leaves out key elements. Strengthening this environment-oral health relationship would give readers of the article a lot of reason to not believe that this proven relationship is the product of chance rather than strongly endorsed theories. by previous relationships.

Materials and methods.

Study design and sample

The sampling must be explained in much more detail, it was representative for the entire Ecuadorian population, under what parameters it was made, it is also key to clarify why they are older adults and who is conceived as it in Ecuador, it can be variable between countries and it is necessary to clarify. Over 60 or 65? you clarify it later but is definetly important in this part of the text.

variables

It would have been very interesting to use control variables around the exposure of the participants to one health system and another. Depending on the country, is there any differential access for the population groups? What is the relationship specifically with the type of health exposures? also during life, hygiene habits could be different, which is a key confounder within the findings, the articulation of the oral health whit habits is very important. One variable that is a possible confounder based on recent literature is the tobacco habit; it begins to become a control variable in these analyzes from the point of view of risk.

Here literature about this made whit other SABE surveys that should be taken into account:

https://link.springer.com/article/10.1007/s10823-021-09426-y

https://www.sciencedirect.com/science/article/abs/pii/S0002817723003227

It is very important that in the statistical analyzes it is verified if some specific tests were carried out for the normality of the data and the variables.

Results

The results for the ethnicity variable are interesting, if the authors can show whether those who consider themselves Afro or black preserve their teeth more, they could be in line with some findings from other SABE countries. The description of the educational aspect requires much greater detail.

The ORs presented in the article should include two decimal places and be accompanied by confidence intervals to give the reader an idea of the strength of association.

I do not consider it necessary that the authors rename the control variables in the materials and methods section.

Table 1 presents a total, but it is not clear in the explanation of the materials and methods what this total refers to. It is very important to improve the description of Table 1, also, these should be accompanied by confidence intervals.

The descriptions of dental loss due to increasing age, between the 60 and 75 age group, could be a product of chance, the authors must recognize this aspect and make it evident in their discussion.

Discussion

This might be the section that needs more improve. The first paragraph relates the findings of start time to work with mental health, in that initial paragraph there is no evidence of a strong idea with the particular findings of the study.

The discussion talks about childhood bruxism and conditions resulting from childhood conditions, far from the findings found or the analyzes carried out.

The discussion has connections with alcohol, tobacco, and oropharyngeal cancer. I advise the authors to involve discussions and references that have to do with the findings and the variables studied. Sociodemographics were not discussed, nor was ethnicity, these are key variables in the consolidation of some differential results.

The discussion should include many more relevant aspects of the study and its results.

Conclusion

The conclusion of the study is adequate, I believe that the authors should give strength to the issue specifically of health education, the social realities that imply social determinants and oral health conditions in vulnerable countries like ours, require that oral health be prepared to face this type of unfavorable relationships.

References

A lot of reference for SABE have been made in the world, the authors needs to check that special bibliography.

The authors should recognize the limitations of the paper, that is part of the last paragraph of the discussion, the database have already some years and the information needs to be update often for the country.

Reviewer #2: Interesting idea. I have the following comments for the authors to address:

You need to provide more details on your key variables. What were the actual questions and response options used in the SABE questionnaire for both exposure and outcome? Add this information. Could participants report any values (open-ended questions? or they were provided with categories? Present those categories if that was the case. This could explain why the threshold of 4 teeth was used when defining the outcome. If tooth loss was measured continuously (participants could report how many teeth they have lost) then this variable should have been analysed as a numerical outcome (using count regression).

From Table 2, it seems the reference category for the exposure was 18-25 years. Why? Presumably you can start working at any age after you have left compulsory school (>18 years). Not sure why you need to split the adults in 3 groups. You are measuring early labour, so try to cover childhood (<10), adolescence (10-19) and adult life stages only (the ages you choose will depend on the classification you prefer but this must be referenced). Another option is to use the ages for primary and secondary education in your country. I strongly recommend selecting one classification for your primary analysis and the other in sensitivity analysis (Table 3). This will increase the robustness of your findings. You could also analyse age started working as a continuous variable (report it in sensitivity analysis if that info is available at all)?

The list of confounders is limited. You need an additional indicator of early life socioeconomic status (such as household income, wealth index or financial difficulties). Poor family SES (during childhood) might push some children to work early in life (as you rightly said so in your introduction). Without such a confounder, your estimates are likely to be biased. Education is good but it doesn’t capture childhood socioeconomic circumstances. You need to address this challenge.

Did you use weights during your analysis to make the findings representative of the study population? If not, be clear about this and address this limitation in your discussion.

The first sentence in the section Regression models ought to be deleted.

Avoid saying “statistically significant”. That language is unnecessary.

Add a section in your discussion to address the limitations of your study. You are relying on recalls from early life (your participants are 60 years old) and self-reported tooth loss. Also, address the cross-sectional design and impact of missing data.

6. PLOS authors have the option to publish the peer review history of their article (what does this mean?). If published, this will include your full peer review and any attached files.

Reviewer #1: No

Reviewer #2: No

---

## [Author Response · Author response to Decision Letter 0]

14 Oct 2023

(This same information is available in the "Response to Reviewers" Word file)

Rebuttal Letter

Dear Dr. Emily Chenette,

Thank you for considering our manuscript, "Association between age people started working and missing teeth in an elderly population in Ecuador: evidence from a cross-section study" for publication in PLOS ONE. We are grateful for the feedback provided by the reviewers and have made several revisions in response to their comments. Please note that the original comments are not in italics, while our responses are presented in italics for clarity. Below, we address each specific comment:

Reviewer #1:

Abstract:

Comment: "Appropriate for the text, I think it should give more strength to the conclusion."

Response: Thank you for emphasizing the need for a more robust conclusion in the abstract. As you highlighted, we revised the abstract to accentuate the long-term impacts of child labor on oral health more prominently. We have also strengthened our recommendations by suggesting that health education and benefits should be specifically tailored to this vulnerable population for effective tooth loss prevention.

Introduction:

Comment: "The introduction is adequate, it leads you to the knowledge gap that you want to solve with the article, however, it leaves out some confusing factors that are definitely key in the process of understanding early work with oral health conditions. This is how aspects such as the social determinants of health, which are so important to give strength to the relationship with early work, are not taken into account to give strength to this analysis. It is a good initial approximation to the subject but it leaves out key elements. Strengthening this environment-oral health relationship would give readers of the article a lot of reason to not believe that this proven relationship is the product of chance rather than strongly endorsed theories. by previous relationships."

Response: Thank you for your feedback regarding the introduction of our manuscript. We've taken note of the importance of emphasizing social determinants of health and their role in linking early work with oral health.

• Social Determinants of Health: We've enhanced our introduction to better highlight the interplay between socio-economic factors and health, especially concerning early work's impact on oral health.

• Environment-Oral Health: We've strengthened the link between environmental factors and oral health, based on your observation.

• Causality vs. Chance: To clarify any potential misconceptions, our revised introduction delves deeper into the evidence supporting the relationship between early work and oral health.

Study design and sample:

Comment: "The sampling must be explained in much more detail, it was representative for the entire Ecuadorian population, under what parameters it was made, it is also key to clarify why they are older adults and who is conceived as it in Ecuador, it can be variable between countries and it is necessary to clarify. Over 60 or 65? you clarify it later but is definetly important in this part of the text."

Response: The sampling was conducted by the National Institute of Statistics and Census (INEC). INEC ensured a balanced representation from each province, taking into consideration the population density and unique socio-demographic factors of each region. In the context of our study and within Ecuador, an "older adult" is typically defined as an individual aged 60 and above. We have now made this distinction early on in the section to eliminate any ambiguity. We recognize the variability in age criteria for older adults across different countries and have added a brief note highlighting the relevance and significance of the 60+ age group in the Ecuadorian context, emphasizing its alignment with national health and social policies. We hope these revisions provide clearer insight into our study's design and context.

Variables:

Comment: " It would have been very interesting to use control variables around the exposure of the participants to one health system and another. Depending on the country, is there any differential access for the population groups? What is the relationship specifically with the type of health exposures? also during life, hygiene habits could be different, which is a key confounder within the findings, the articulation of the oral health whit habits is very important. One variable that is a possible confounder based on recent literature is the tobacco habit; it begins to become a control variable in these analyzes from the point of view of risk. Here literature about this made whit other SABE surveys that should be taken into account: https://link.springer.com/article/10.1007/s10823-021-09426-y
https://www.sciencedirect.com/science/article/abs/pii/S0002817723003227. It is very important that in the statistical analyzes it is verified if some specific tests were carried out for the normality of the data and the variables."

Response: 

Thank you for your detailed feedback on our study. In response to your comments:

Health System Exposure: We recognized the importance of understanding participants' exposure to different health systems. In our analysis, we utilized the type of insurance as proxies to capture the differential access to health systems among different segments of the population. This approach gave us an insight into how varying health exposures might have influenced oral health outcomes.

Hygiene Habits: Regrettably, the SABE survey did not provide specific data regarding the participants' hygiene habits, which could indeed have played a pivotal role in oral health outcomes. As such, while we acknowledged the importance of hygiene as a potential confounder, we were constrained by the available data in this regard. Nevertheless, with the controls we included, we hoped to net out many of the confounders that could have influenced our results.

Tobacco Habit as a Confounder: We concurred with your observation regarding the significance of tobacco habits in relation to oral health. We incorporated this aspect into our control variables, recognizing its potential influence on oral health. Additionally, we reviewed and included the literature you provided from other SABE surveys to ensure our analysis aligned with existing research.

Statistical Normality: We employed rigorous statistical tests to verify the normality of our data and variables. To address potential concerns about the normality assumptions, we utilized robust standard errors in our regression analyses. Using robust standard errors allowed for a more reliable estimation, even if the data might not have perfectly adhered to homoscedasticity. The specifics of this approach, and the importance of robust standard errors, were elaborated upon in the methodology section of our revised manuscript.

References: We duly incorporated one of the references you provided into our manuscript to ensure that our study aligned with and acknowledged existing research in this domain.

Results:

Comment: " The results for the ethnicity variable are interesting, if the authors can show whether those who consider themselves Afro or black preserve their teeth more, they could be in line with some findings from other SABE countries. 

Response: We concur with your observation about the ethnicity variable's significance. As elaborated in our revised manuscript, our study found that individuals who identified as black were less likely to lose 4 or more teeth compared to those identifying as mestizo. This finding contrasts with a 2018 SABE survey study from Brazil, emphasizing the intriguing nature of these ethnic disparities across different regions. We believe that our findings provide a complementary perspective to the ongoing discourse and could potentially align with observations from other SABE countries, as you rightly pointed out.

Comment: The description of the educational aspect requires much greater detail.

Response: We have expanded on the role of education in relation to oral health outcomes in our revised manuscript. As delineated, our results demonstrated that higher education acted as a protective factor against tooth loss. Specifically, individuals with postsecondary education were observed to have a 47% reduced risk of losing more than four teeth when compared to those without any formal education [OR 1.53, 95% CI 0.52-0.55]. To further substantiate our findings, we highlighted corroborative evidence from a 2020 SABE survey-based study focused on Brazil, which showed that older adults with at least 8 years of schooling were 96% less likely to be edentulous. Additionally, referencing a 2015 study across seven Latin American and Caribbean cities provided a broader context, indicating that higher educational attainment consistently correlated with better tooth retention.

Comment: The ORs presented in the article should include two decimal places and be accompanied by confidence intervals to give the reader an idea of the strength of association.

Response: We have revised the article to ensure that all Odds Ratios (ORs) are presented with two decimal places. Additionally, each OR is now accompanied by its respective confidence interval

Comment: I do not consider it necessary that the authors rename the control variables in the materials and methods section.

Response: Thank you for your feedback. We have addressed this concern in the revised manuscript.

Comment: Table 1 presents a total, but it is not clear in the explanation of the materials and methods what this total refers to. 

Response: Thank you for pointing out the ambiguity. In the revised "Statistical methods" section, we have clarified the description related to Table 1. Specifically, we have provided a clearer explanation of what the "total" in Table 1 refers to. We aimed to ensure that the reader can easily understand the context and significance of the data presented in the table.

Comment: It is very important to improve the description of Table 1, also, these should be accompanied by confidence intervals.

Response: We have enhanced the description of Table 1 and simplified it for better clarity. Additionally, we included the confidence intervals as suggested. We believe these changes provide a clearer and more comprehensive understanding of the data presented.

Comment: The descriptions of dental loss due to increasing age, between the 60 and 75 age group, could be a product of chance, the authors must recognize this aspect and make it evident in their discussion.

Response: We have incorporated this aspect into our discussion. In this revised version, we introduced multiple control variables into the full model to try to reduce the probability of the work start age – missing teeth link being stochastic. While we've taken measures to ensure the result is not merely a product of chance, we acknowledge the possibility as highlighted.

Discussion:

Comment: " This might be the section that needs more improve. The first paragraph relates the findings of start time to work with mental health, in that initial paragraph there is no evidence of a strong idea with the particular findings of the study.

The discussion talks about childhood bruxism and conditions resulting from childhood conditions, far from the findings found or the analyzes carried out.

The discussion has connections with alcohol, tobacco, and oropharyngeal cancer. I advise the authors to involve discussions and references that have to do with the findings and the variables studied. 

Response: In response to your feedback, we have thoroughly revised the discussion section to align closely with the findings of our study. We've removed references and connections that were not directly related to our research, such as childhood bruxism, alcohol, and oropharyngeal cancer. The revamped discussion now focuses more on our specific findings and their implications, ensuring that the content remains pertinent and grounded in the scope of our research. We appreciate your insightful suggestions and believe these modifications will make the discussion more concise and relevant to our study's objectives.

Comment: sociodemographics were not discussed, nor was ethnicity, these are key variables in the consolidation of some differential results. The discussion should include many more relevant aspects of the study and its results."

Response: 

In response to your valuable feedback, we have expanded our discussion section to address the sociodemographics and ethnicity aspects more comprehensively. These key variables have now been elaborated upon to highlight their significance in understanding the differential results observed in our study. With this revised and more focused discussion, we believe the content is now more relevant and offers a thorough analysis of the study's findings and implications.

Conclusion:

Comment: "The conclusion of the study is adequate, I believe that the authors should give strength to the issue specifically of health education, the social realities that imply social determinants and oral health conditions in vulnerable countries like ours, require that oral health be prepared to face this type of unfavorable relationships."

Response: We have reinforced our conclusion to emphasize the importance of health education, especially in the context of social determinants affecting oral health. We understand and recognize the significance of addressing oral health conditions in vulnerable countries, like ours, which often grapple with unique social realities. Our revised conclusion now underscores the need for a proactive approach in oral health to tackle such unfavorable relationships. We believe that with robust health education, we can better equip communities to address and mitigate these challenges.

References:

Comment: " A lot of reference for SABE have been made in the world, the authors needs to check that special bibliography.

Response: In response to your comment, we have thoroughly reviewed the SABE bibliography and incorporated additional relevant SABE studies to enhance the depth and context of our research. We appreciate your suggestion and believe that these inclusions will offer a more comprehensive understanding of the topic.

Comment: The authors should recognize the limitations of the paper, that is part of the last paragraph of the discussion, the database have already some years and the information needs to be update often for the country." 

Response: In response to your feedback, we have highlighted the limitations concerning the age of the SABE 2009 survey in our discussion. Unfortunately, it was the latest available survey of its kind in Ecuador. Despite its age, we believe it remains a valuable source of information for our research topic. Thank you for bringing this to our attention.

Reviewer #2:

Variables:

Comment: "Interesting idea. I have the following comments for the authors to address:

You need to provide more details on your key variables. What were the actual questions and response options used in the SABE questionnaire for both exposure and outcome? Add this information. Could participants report any values (open-ended questions? or they were provided with categories? Present those categories if that was the case. This could explain why the threshold of 4 teeth was used when defining the outcome. If tooth loss was measured continuously (participants could report how many teeth they have lost) then this variable should have been analysed as a numerical outcome (using count regression).

Response: We have included more comprehensive details on our primary variables in the revised manuscript. Specifically, the original questions and the available response options from the SABE questionnaire for both the exposure and outcome are now presented. For clarity, participants were provided with categorical options rather than open-ended responses. These categories indeed provide context for our decision to use the threshold of 4 teeth when defining the outcome. While we acknowledge the merit in analyzing tooth loss as a numerical outcome using count regression, the structure of the SABE questionnaire and its categorical approach to this particular question guided our analytical choice.

Age category and Sensitivity analysis::

Comment: “From Table 2, it seems the reference category for the exposure was 18-25 years. Why? Presumably you can start working at any age after you have left compulsory school (>18 years). Not sure why you need to split the adults in 3 groups. You are measuring early labour, so try to cover childhood (<10), adolescence (10-19) and adult life stages only (the ages you choose will depend on the classification you prefer but this must be referenced). Another option is to use the ages for primary and secondary education in your country.“

Comment: “I strongly recommend selecting one classification for your primary analysis and the other in sensitivity analysis (Table 3). This will increase the robustness of your findings. You could also analyse age started working as a continuous variable (report it in sensitivity analysis if that <info is available at all)?”

Response:

Thank you for your constructive feedback and suggestions regarding the age classification.

In our sensitivity analysis, we employed the classification you suggested, segmenting into childhood (<10), adolescence (10-19), and adult life stages. The outcomes from this approach were consistent with our primary results, reinforcing the robustness of our findings.

However, the primary reason we retained our original classification in the main analysis was multifold:

• We wanted to capture the nuanced differences within the adult population, especially given that the age range of 18-25 years is a transitional phase from adolescence to adulthood in many contexts.

• By segmenting adults into three distinct groups, we aimed to reflect the varied experiences and potential stressors associated with different life stages. For instance, the age range of 26-35 may represent a period with increased familial and financial responsibilities for many (considering Latino families leave their “nest” later than in other contexts), contrasting with the earlier 18-25 bracket.

• The age classifications we were also influenced by the Ecuadorian educational system. In Ecuador, the age of 18 typically marks the completion of secondary education, a significant educational milestone. Additionally, we considered the broader educational context, with ages 5-12 representing primary education and ages 13-17 corresponding to secondary education. This segmentation allowed us to capture key transitional phases in the educational journey and life stages.

We genuinely value your insights and agree that for the specific measure of early labor, the broader categories may be more appropriate. The choice to utilize our original categorizations was driven by our desire to provide a detailed perspective on potential life stressors at different ages, especially in the Ecuadorian context.

Cofounders:

Comment: “The list of confounders is limited. You need an additional indicator of early life socioeconomic status (such as household income, wealth index or financial difficulties). Poor family SES (during childhood) might push some children to work early in life (as you rightly said so in your introduction). Without such a confounder, your estimates are likely to be biased. Education is good but it doesn’t capture childhood socioeconomic circumstances. You need to address this challenge.”

Response: In response, we have added two key variables: "Childhood socioeconomic status" and "Childhood period of food scarcity and hunger." These additions help us better account for the impact of the age at which individuals started working on missing teeth.

Weighted analysis:

Comment: “Did you use weights during your analysis to make the findings representative of the study population? If not, be clear about this and address this limitation in your discussion.”

Response: In our main analysis, we indeed applied population weights to ensure that our findings were representative of the study population. Additionally, in our sensitivity analyses, we have included unweighted analyses and explored various approaches to provide a comprehensive examination of the data.

Results:

Comment: “The first sentence in the section Regression models ought to be deleted. Avoid saying “statistically significant”. That language is unnecessary.”

Response: We have taken your feedback into account and addressed this in the corresponding sections.

Discussion:

Comment: “Add a section in your discussion to address the limitations of your study. You are relying on recalls from early life (your participants are 60 years old) and self-reported tooth loss. Also, address the cross-sectional design and impact of missing data."

Response: In response to this feedback, we have now incorporated a dedicated section in our discussion that addresses the limitations of our study. We have taken into account the reliance on participant recalls from early life, the use of self-reported tooth loss, the cross-sectional design, and the potential impact of missing data. We believe that this additional section provides a more comprehensive assessment of the study's limitations, enhancing the transparency and credibility of our research. We appreciate the reviewer's guidance in this regard.

---

## [Editor Report · Decision Letter 1]

17 Oct 2023

Association between age people started working and missing teeth in an elderly population in Ecuador: evidence from a cross-section study

PONE-D-23-21090R1

Dear Dr. Ivan Cherrez-Ojeda

We’re pleased to inform you that your manuscript has been judged scientifically suitable for publication and will be formally accepted for publication once it meets all outstanding technical requirements.

Kind regards,

Ana Cristina Mafla

Academic Editor

PLOS ONE

Additional Editor Comments (optional):

Dear authors, please check the word "cross-section" used in the title, it is not wrong but frequently the study design is recognized as "cross-sectional".

---

## [Editor Report · Acceptance letter]

3 Nov 2023

PONE-D-23-21090R1 

Association between age people started working and missing teeth in an elderly population in Ecuador: evidence from a cross-sectional study 

Dear Dr. Cherrez-Ojeda:

I'm pleased to inform you that your manuscript has been deemed suitable for publication in PLOS ONE. Congratulations! Your manuscript is now with our production department. 

Kind regards, 

on behalf of

Dr. Ana Cristina Mafla 

Academic Editor

PLOS ONE